# Health Risk Assessment of Banned Veterinary Drugs and Quinolone Residues in Shrimp through Liquid Chromatography–Tandem Mass Spectrometry

**Ming-Yang Tsai [1,2,†], Chuen-Fu Lin [3,†], Wei-Cheng Yang [4], Chien-Teng Lin [3], Kuo-Hsiang Hung [2] and Geng-Ruei Chang [3,*]**

[1]  Animal Industry Division, Livestock Research Institute, Council of Agriculture, Executive Yuan, Tainan 71246, Taiwan; mytsai@mail.tlri.gov.tw

[2]  Graduate Institute of Bioresources, National Pingtung University of Science and Technology, Pingtung 91201, Taiwan; khhung424@mail.npust.edu.tw

[3]  Department of Veterinary Medicine, National Chiayi University, Chiayi 60054, Taiwan; cflin@mail.ncyu.edu.tw (C.-F.L.); vet540423@gmail.com (C.-T.L.)

[4]  Department of Veterinary Medicine, School of Veterinary Medicine, National Taiwan University, Taipei 10617, Taiwan; yangweicheng@ntu.edu.tw

*  Correspondence: grchang@mail.ncyu.edu.tw; Tel.: +886-5-2732959; Fax: +886-5-2732917

†  These authors contributed equally to this work.

**Abstract:** The presence of antibiotic residues in seafood and their effect on public health constitute a matter of concern for consumers worldwide. Antibiotic residues can have adverse effects on both humans and animals, especially residues of banned veterinary drugs. In this study, we applied a validated method to analyze veterinary drug residues in shrimp, including the levels of banned chloramphenicol, malachite green, leucomalachite green, and four nitrofuran metabolites as well as thiamphenicol, florfenicol, and five quinolones, which have no recommended maximum residual levels in shrimp tissues in Taiwan. We collected 53 samples of whiteleg, grass, or giant river shrimp from Taiwanese aquafarms and production areas from July 2016 to December 2017. We found 0.31 ng/g of a chloramphenicol in one grass shrimp, 5.62 ng/g of enrofloxacin in one whiteleg shrimp, 1.52 ng/g of flumequine in one whiteleg shrimp, and 1.01 ng/g of flumequine in one giant river shrimp, indicating that 7.55% of the samples contained veterinary drug residues. We evaluated the health risk by deriving the estimated daily intake (EDI). The quinolone residue EDI was below 1.0% of the acceptable daily intake recommended by the United Nations Food and Agriculture Organization and World Health Organization. The risk was thus discovered to be negligible, indicating no immediate health risk associated with shrimp consumption. The present findings can serve as a reference regarding food safety and in monitoring of the veterinary drug residues present in aquatic organisms. Continual monitoring of residues in shrimp is critical for further assessment of possible effects on human health.

**Keywords:** veterinary drug; residue; shrimp; mass spectrometry; risk assessment

## 1. Introduction

Taiwan has a geographic location and environment conducive to aquaculture development. Aquaculture in Taiwan has a long history of more than three centuries, and it has rapidly expanded, diversified, intensified, and technologically advanced from 1960 to the 1990s [1]. Despite Taiwan's land and water resource limitations, it is one of the major aquaculture producers in the world; therefore, it was once called the "kingdom of aquaculture" [2]. Until now, over 35 major and candidate species have been cultured for commercial purposes [3]. The average revenue of Taiwan from

aquaculture reached US $11 billion between 2010 and 2015 [3]. Moreover, the annual aquaculture production was approximately 300,000 t during the 2010s. Specifically, shrimp culture production has been notable because Taiwan's government has strongly supported aquaculture since the 1980s [4], in particular, aquaculture of shrimp—including whiteleg shrimp (*Litopenaeus vannamei*), grass shrimp (*Penaeus monodon*), giant river shrimp (*Macrobrachium rosenbergii*), sand shrimp (*Metapenaeus ensis*), and kuruma shrimp (*Penaeus japonicus*). Moreover, the average revenue from the shrimp aquaculture industry reached US $3000 million between 2010 and 2016.

Intensive and large-scale breeding is preferred by aquaculture farmers in Taiwan because the farmers are limited to land use [5]. Moreover, their farms are generally situated near residential and agricultural areas, which makes biological control difficult [6]. The cultured species became more susceptible to bacterial, viral, parasitical, and fungal infections, necessitating the use of various veterinary drugs for the prevention and treatment of these infections. However, when such drugs are heavily employed in aquaculture, aquaculture products may contain drug residues, potentially exposing the consumers of the products to these residues. This has an inadvertent ecological impact and raises health concerns such as increased risk of allergies, carcinogen exposure, and the development of bacteria resistant to antibiotics [5]. Because of their benthic feeding behavior, shrimp can be used to indicate the levels of chemicals in aquatic environments [7]. Analyzing the amounts of pollutants in shrimp indicates the environmental levels of veterinary drugs and the extent to which drugs are transferred through the trophic chain.

In Taiwan, shrimp are the crustaceans cultured most extensively in land-based ponds [3], and in the inner regions of the island, shrimp are generally bred in a mix with other aquatic products. Typically, veterinary drugs are used to prevent or treat diseases in nonshrimp targets, which pollutes water and soil environments while contaminating shrimp. In this regard, shrimp products have played a crucial role in seafood safety. Residues of veterinary drugs in shrimp are a crucial concern regarding public health, especially when the residues are of banned chemicals that have been employed illegally. Therefore, in this study, we detected in shrimp samples the levels of the following banned veterinary compounds in Taiwan: leucomalachite green (LMG), malachite green (MG), nitrofuran metabolites, and chloramphenicol. These compounds' maximum residue limits (MRLs) in shrimp have not been established by the Taiwan Food and Drug Administration (TFDA). In addition, we detected florfenicol, thiamphenicol, and quinolone residues, including danofloxacin, difloxacin, enrofloxacin, flumequine, and sarafloxacin, in shrimp. The TFDA defined the MRL levels of these quinolones in livestock and chicken in 2018; however, the use of these compounds in the cultivation of shrimp is banned. The present study's detection of the residues of these compounds in shrimp thus indicates the degree of legal compliance regarding the use of these products. In addition, the seafood-consumption-based estimated daily intake (EDI) of these contaminants was determined to identify the effects of exposure to these veterinary drugs on the health of the Taiwanese public. Results were applied for assessing the risk of exposure to veterinary drugs among consumers in Taiwan. The findings of this study are useful when conducting evaluations of seafood safety and may be used as a reference among health authorities for establishing regulations.

## 2. Materials and Methods

### 2.1. Samples

Shrimp samples were obtained between July 2016 and December 2017 from aquafarms in the principal production areas (Yunlin, Chiayi, Tainan, Kaohsiung, and Pingtung). In total, 53 samples (23 whiteleg, 16 grass, and 14 giant river shrimps) were collected. These shrimps are bred on a large scale in Taiwan [3]. We removed, homogenized, and stored the soft tissues of all shrimp samples at −20 °C until they were analyzed.

### 2.2. Chemicals and Reagents

Analytical compounds included veterinary drugs, namely chloramphenicol (98.6%), thiamphenicol (98.5%), florfenicol (98%), MG (98.0%), and LMG (99.0%), purchased from Dr. Ehrenstorfer GmbH (Ausburg, Germany). Nitrofuran metabolites, namely 5-methylmorpholino-3-amino-2-oxazolidinone (AMOZ, 99.9%), 3-amino-2-oxazolidinone (AOZ, 99.7%), and 1-aminohydantoin hydrochloride (99.9%), were obtained from Sigma-Aldrich (St. Louis, MO, USA). Semicarbazide hydrochloride (99.5%) was obtained from Chem Service Inc. (West Chester, PA, USA). Danofloxacin (98.0%), difloxacin (98.0%), enrofloxacin (98.0%), flumequine (98.0%), and sarafloxacin (95.0%) were obtained from Sigma-Aldrich. In addition, stable isotopically labeled internal standards, AMOZ-D5 (99.0%) and AOZ-D4 (99.0%), were purchased from Dr. Ehrenstorfer GmbH. Other internal standards, namely SC-13C15N2 (99%), MG-D5 picrate (99.9%), and LMG-D5 (99.8%), were purchased from Sigma-Aldrich.

Chromatography-grade acetonitrile (ACN), acetone, ammonium acetate, dipotassium phosphate, ethyl acetate (EtOAc), formic acid (FA), hydrochloric acid (HCl), methanol (MeOH), n-hexane, and sodium hydroxide (NaOH) were purchased from Merck (Darmstadt, Germany). Reagent-grade 2-nitrobenzaldehyde (2-NBA), sodium chloride (NaCl), and *N*,*N*,*N*′,*N*′-tetramethyl-1,4-phenylenediamine dihydrochloride (TMPD) were purchased from Sigma-Aldrich.

### 2.3. Instruments and Apparatus

A vortex mixer (type 37600 mixer, Barnstead/Thermolyne, Dubuque, IA, USA), a centrifuge (Allegra X-22R, Beckman Coulter, Fullerton, CA, USA), a nitrogen evaporator (N-Evap-111, Organomation Associates Inc., Berlin, Germany), and a nitrogen generator (Model 05B, System Instruments Co., Tokyo, Japan) were used to prepare samples. The liquid chromatography–tandem mass spectrometry (LC/MS–MS) apparatus comprised an LC system (Agilent Technologies 1200, Agilent Technologies, Palo Alto, CA, USA) and a mass spectrometer (ABI 4000 QTRAP, Applied Biosystems, Foster City, CA, USA). To determine the levels of residues of chloramphenicol classes, nitrofuran metabolites, and quinolone classes in samples, chromatographic separation was performed in an analytical column (Chromolith Performance RP-18e, 100 mm × 3 mm, Merck, Darmstadt, Germany) and a guard column (Chromolith Guard Column RP-18e, 5 mm × 4.6 mm, Merck). In addition, MG and LMG were separated using a Purospher STAR RP-18 endcapped analytical column (100 mm × 2.1 mm × 2 μm, Merck) and Purospher Star RP-18 endcapped guard column (4 mm × 4 mm × 5 μm, Merck).

### 2.4. Analysis of LC/MS–MS Conditions

An injection volume of 10 μL was used for determining the levels of chloramphenicol classes, MG, LMG, and quinolone classes and 20 μL for nitrofuran metabolites. Chloramphenicol, thiamphenicol, and florfenicol levels were analyzed through gradient elution by using the A1 eluent (0.1% MeOH) and B1 eluent (100% MeOH). A mobile phase gradient was started at 40% B1 for 1 min at a flow rate of 0.5 mL/min, linearly increased to 90% B1 at 4 min, and subsequently maintained constant until 6 min. Thereafter, it was changed to 40% B1 after 6.1 min and maintained constant until 9 min. Danofloxacin, difloxacin, enrofloxacin, flumequine, and sarafloxacin levels were analyzed through gradient elution by using the A2 eluent (0.1% FA) and B2 eluent (100% MeOH). A mobile phase gradient was started at 10% B1 for 1 min at a flow rate of 0.5 mL/min, linearly increased to 95% B1 at 4 min, and subsequently maintained constant until 8 min. Thereafter, it was changed to 10% B1 after 8.1 min and maintained constant until 9 min. The A3 eluent (0.005 M ammonium acetate in 0.1% FA) and B2 eluent (0.005 M ammonium acetate in MeOH) were used as the mobile phase for nitrofuran metabolite analysis. The mobile phase gradient was started with 30% B2 at a flow rate of 0.3 mL/min, increased linearly to 95% B2 in 4 min, further maintained until 6 min, subsequently changed to 30% B2 after 7 min, and maintained constant until 10 min. The A4 eluent (0.1% FA) and B4 eluent (MeOH) were used as the mobile phase for MG and LMG analyses, respectively. These dyes were separated following the gradient program. The mobile phase gradient was started with 10% B4 for 1 min at a flow rate

of 0.5 mL/min, increased from 10% to 95% B4 in 4 min, and maintained constant until 8 min. Finally, B4 was changed to 10% after 8.1 min and maintained constant until 11 min. The MS source conditions in ABI 4000 QTRAP were as follows: ion spray voltage of 4.5–5.5 kV, curtain gas of 15 psi, nebulizer gas of 50 psi, auxiliary gas of 60 psi, and source temperature of 50 °C. MS/MS experiments were conducted in multiple reaction monitoring modes (MRMs) for simultaneous detection of all targets, with two precursor-to-product ion transitions monitored for each analyte. The mass spectrometer was set to detect negative and positive ESI interface modes for chloramphenicol and other veterinary drugs, respectively. Supplementary Table S1 lists the retention times and the precursor and corresponding product ions obtained through MRM detection in LC-amenable veterinary analytes. The dwell time for each MRM transition was set at 5 ms. Analyst software (version 1.4, Applied Biosystems, Foster City, CA, USA) was used for instrument control and data acquisition.

### 2.5. Preparing the Standard Solutions

In volumetric flasks, stock solutions were prepared that contained pesticide standards or individual veterinary drugs by dissolving each analyte (100 mg) in 100 mL of—depending on the solubility of the analytes—ACN, acetone, or MeOH. All types of stock solution were combined and diluted to 1 mg/L to obtain a working standard mixture. We stored all solutions at −20 °C, and before use, a solution was allowed to adjust to room temperature. With these working standard solutions, serial dilution was performed to prepare a series of calibration standards (dilution range 0.5–500 ng/mL).

### 2.6. Extraction Procedure and Analysis

To detect residues of chloramphenicol classes, MG, LMG, and quinolone classes, we extracted and cleaned each shrimp sample by using a modification of the veterinary drug residue analysis technique reported by Chang et al. [5] and Smith et al. [8] for aquatic products. Briefly, 2 g of sample was weighed in a propylene centrifuge tube (volume 50 mL) and transferred to a homogenizer containing 100 µL of internal standards (100 ng/mL), 50 µL of TMPD, and 10 mL of ACN. Then, we added 5 mL of n-hexane saturated with CAN to the homogenate, which was shaken in a vortex mixer for 5 min, followed by centrifugation at 4500 rpm for 10 min. We aspirated and subsequently discarded the hexane layer. The ACN extraction layer was collected and dried at 40 °C in a nitrogen evaporator. We re-extracted the remaining tissue pellets using 10 mL of ACN and 5 mL of ACN-saturated n-hexane and then centrifuged them. The first extract was combined with the ACN layer. Subsequently, we evaporated the combined extracts to dryness at 0.5 mL. An additional 0.5 mL of ultrapure water was added, after which the vortex was mixed and then sonicated for 1 min. The reconstituted extracts underwent centrifugation at 4500 rpm for 5 min. Finally, a 0.2 µm polyvinylidene fluoride filter (Whatman, Maidstone, UK) was employed to filter the supernatant layer, and the filtrate was transferred to an autosampler vial prior to being injected into the chromatographic system.

Nitrofuran metabolite extraction from samples was performed through the execution of a TFDA-procedure-based method [9]. Briefly, in a centrifuge tube measuring 50 mL in volume, we fortified 2 g of a sample with 100 µL of internal standards (100 ng/mL), followed by sequentially adding 0.125 M HCl (9 mL) and 50 mM 2-NBA in MeOH (400 µL). Samples were vortex mixed (1 min), followed by overnight incubation (16 h, 37 °C) with gentle shaking in a water bath. In order to neutralize the samples, we added 0.8 M NaOH (1 mL) and 0.1 M dipotassium phosphate buffer (1 mL), and we adjusted the reaction mixture to pH 7.1–7.5. The mixture underwent 1 min of vigorous vortex mixing and was then centrifuged at 3500 rpm for 5 min. After the collection of the supernatant, the remaining tissue pellet was re-extracted using ultrapure water (3 mL), as described earlier in the text, and centrifuged again. The combined extracts were re-extracted using 0.5 g of NaCl and 12 mL of EtOAc with vortex shaking of the samples for 1 min. The reconstituted extracts were again centrifuged for 5 min at 3500 rpm. The solvent was evaporated at 40 °C in a nitrogen evaporator. We reconstituted the resultant dry extract in 1 mL of 50% MeOH, after which it was vortex mixed for 1 min. Subsequently, 1 mL of n-hexane was added to the extracts, which then underwent centrifugation again, as described

earlier. The lower layer was collected and filtered (0.2 μm filter membrane). The filtrate was placed in an autosampler vial prior to analysis.

### 2.7. Assurance and Validation of Quality

We validated the proposed method by calculating the recovery, linearity range, repeatability, and limits of quantification (LOQs) [10,11]. For determining the recovery and repeatability, we spiked blank samples in triplicate by using the following standard mixture of analytes at two concentrations (low and high levels): 1 and 5 ng/g for determination of chloramphenicol classes, dyes, and nitrofuran metabolites; and 5 and 25 ng/g for determination of quinolone classes. Extraction and treatment of the samples followed a previously reported protocol [2,8,9]. The aforementioned recovery validation method was employed to determine the method's repeatability, which was calculated as the percentage of the relative standard deviation (RSD%). The recovery and repeatability (expressed as the percentage of relative standard deviation) of veterinary drugs ranged from 88.67% to 92.35% (repeatability range: 3.79–9.67%) for chloramphenicol classes, 75.21% to 103.31% (repeatability range: 6.72–14.58%) for quinolone classes, 98.81% to 100.31% (repeatability range: 3.58–8.13%) for MG and LMG, and 99.29% to 100.52% (repeatability range: 0.98–5.58%) for nitrofuran metabolites in shrimp samples (Supplementary Table S2). Matrix-matched calibration executed through the use of blank sample extracts and addition of the corresponding amount of working solution (with target compounds at a concentration of 0.5–500 ng/mL) was performed to evaluate the linearity. The calibration curves obtained had high linearity and reproducibility, with the analytical matrix-matched calibration achieving favorable correlation coefficients ($R^2 > 0.990$). The LOQs were defined as being the concentrations of analyte that yielded peak signals 3× and 10× the intensity of background noise from the chromatogram. The florfenicol, thiamphenicol, chloramphenicol, LMG, MG, and nitrofuran metabolite LOQ was 0.25 ng/mL in shrimp samples. Compared with these chemicals, the LOQ of other veterinary drugs, including danofloxacin, difloxacin, enrofloxacin, flumequine, and sarafloxacin, was 1 ng/g (Supplementary Table S2); concentrations lower than these LOQs indicated that the chemicals and drugs were considered undetectable.

### 2.8. EDI

To assess the degree to which people are exposed to veterinary drug residues in shrimp, we estimated the EDI from the residual veterinary drug concentrations. The acceptable daily intakes (ADIs) established by the World Health Organization (WHO) and Food and Agriculture Organization of the United Nations (FAO) were employed as points of comparison. The following equation was used to calculate the EDI: EDI (ng/kg/day) = (daily fish consumption [g/day]) × (mean veterinary drug concentration [ng/g])/(human body weight [kg]) [6]. Data regarding Taiwanese citizens' daily seafood consumption (96.9 g for men and 74.2 g for women) were collected from the National Nutrition and Health Survey conducted by the Ministry of Health and Welfare [12]. We considered the mean Taiwanese body weight to be 60 kg [12]. We determined the maximal EDI from the maximum residue concentrations.

## 3. Results

### 3.1. Detection Rates and Levels of Veterinary Drugs in Shrimp Samples

In total, 23 whiteleg, 16 grass, and 14 giant river shrimp samples were collected. Chloramphenicol was detected in one grass shrimp, enrofloxacin in one whiteleg shrimp, and flumequine in one whiteleg shrimp and one giant river shrimp (Table 1). These detected veterinary drugs are prohibited by the TFDA for use in shrimp. In all shrimp samples, the predominant residue was flumequine at 3.77% (2/53), followed by chloramphenicol at 1.89% (1/53) and enrofloxacin at 1.89% (1/53). Veterinary drugs were detected in 8.70% (2/23), 6.25% (1/16), and 7.14% (1/14) of the whiteleg, grass, and giant river shrimp samples, respectively. The levels of chloramphenicol and enrofloxacin were 0.29 and 5.62 ng/g

in one grass and whiteleg shrimp, respectively. Moreover, flumequine (1.01–1.52 ng/g) was detected in two shrimp samples, namely in one whiteleg and one giant river shrimp. The concentrations of chloramphenicol, enrofloxacin, and flumequine (derived from all samples, including samples with detected and undetected concentrations) were 0.01, 0.11, and 0.05 ng/g, respectively. Overall, 7.55% (4/53) of all shrimp samples contained detectable veterinary drug residues, which indicated the positive ratio of banned residual drugs.

**Table 1.** Detection levels of banned veterinary drugs in various shrimp samples collected between July 2016 and December 2017.

| Shrimp | Surveyed Samples | Violated Targets (No.) | Detected Residues (ng/g) | Average [1] (ng/g, Residues) | Violated Ration [2] (%) |
|---|---|---|---|---|---|
| Whiteleg shrimp | 23 | enrofloxacin (1) flumequine (1) | 5.62 1.52 | 0.24 (enrofloxacin) 0.07 (flumequine) | 3.77 (2/53) |
| Grass shrimp | 16 | chloramphenicol (1) | 0.31 | 0.02 (chloramphenicol) | 1.89 (1/53) |
| Giant river shrimp | 14 | flumequine (1) | 1.01 | 0.07 (flumequine) | 1.89 (1/53) |
| Total | 53 53 53 | chloramphenicol (1) enrofloxacin (1) Flumequine (2) | 0.31 5.62 1.01–1.52 | 0.01 0.11 0.05 | 7.55 (4/53) |

[1] Estimated from all samples, including samples with detected and undetected concentrations. [2] Samples with residual concentrations lower than the LOQ were considered to have undetectable concentrations.

### 3.2. EDIs of Taiwanese Adults for Veterinary Drug Residues in Shrimp Samples

The Joint FAO/WHO Expert Committee on Food Additives (JECFA) determined the inappropriateness of establishing a chloramphenicol ADI [13]. Therefore, we did not estimate the EDI of chloramphenicol residues in shrimp samples. The EDIs calculated from the average enrofloxacin and flumequine levels were, respectively, 0.14 and 0.06 ng/kg body weight/day for women and 0.18 and 0.08 ng/kg body weight/day for men (Table 2). Regarding the veterinary drug residues in food, the ADIs stipulated by the JECFA's Joint Meeting of the FAO/WHO for enrofloxacin and flumequine are 0.002 and 0.03 mg/kg, respectively [14,15]. As detailed in Table 2, the obtained EDIs were considerably lower than the enrofloxacin and flumequine ADIs recommended by the FAO/WHO. For enrofloxacin and flumequine, the EDIs expressed as a percentage of the ADIs were, respectively, 0.01% and 0.0003% for men and 0.01% and 0.0002% for women. Overall, consumption of shrimp lead to a low risk of dieldrin exposure, with the ADIs lower than 1.0% for men and women.

**Table 2.** Estimated dietary intake of quinolone residues in Taiwanese adults.

| OCPs | EDI (ng/kg Body Weight/Day) | | EDI% of ADI | | ADI (FAO/WHO) (mg/kg Body Weight/Day) |
|---|---|---|---|---|---|
| | Male | Female | Male | Female | |
| Enrofloxacin | 0.18 | 0.14 | 0.01 | 0.01 | 0.002 |
| Flumequine | 0.08 | 0.06 | 0.0003 | 0.002 | 0.03 |

## 4. Discussion

In the present study, 14 residual veterinary drugs, namely three chloramphenicol classes (chloramphenicol, florfenicol, and thiamphenicol), five quinolone classes (danofloxacin, difloxacin, enrofloxacin, flumequine, and sarafloxacin), MG, LMG, and four nitrofuran metabolites (AMOZ, AOZ, AH, and SC), were analyzed in 52 shrimp samples collected from aquaculture areas in Taiwan. To validate the presence of these compounds in samples, we evaluated the mean recovery (as a measure of trueness), linearity, sensitivity, and repeatability (as a measure of precision) according to EU guidelines (SANCO/12495/2011) [10]. Because chloramphenicol, nitrofuran metabolites, MG, and LMG are banned from use in edible animals and danofloxacin, difloxacin, enrofloxacin, flumequine, and sarafloxacin are banned from use in decapods, the TFDA does not recommend MRLs in shrimp.

Therefore, the residues of these banned compounds in shrimp were detected and sufficiently indicated the degree of legal compliance regarding the use of these products.

The Commission Decision 2002/657/EC criteria for evaluation of veterinary drug residues in animals and animal products are stipulated on the basis of mass spectrometry at numerous identification points (IPs) [16]. Source conditions were optimized to obtain 1.5 IP from product ions and one IP from precursor ions for each compound. In general, obtaining four IPs at the lowest level is required for analyzing banned compounds. In the present study, veterinary drugs were analyzed in the MRM mode by monitoring three different ions (one precursor and two fragment ions). Using this approach, we achieved four IPs (one IP from a single precursor ion and three IPs from two fragment ions), as mandated by the aforementioned guidelines. Our analysis method successfully identified the residues of MG, LMG, chloramphenicol classes, quinolone classes, and nitrofuran metabolites.

The analytical extraction method for aquatic samples was designed by Smith et al. [8]. In this method, ACN and hexane are used to extract samples for simultaneously screening multiple classes of drug residues, including macrolides, β-lactam antibiotics, dyes, quinolones, tetracyclines, and antimycotic imidazoles. Moreover, other extraction methods have been reported for analyzing chloramphenicol [17], MG, and LMG [18]. In addition, we applied this method for the extraction of chloramphenicol, MG, and LMG residues in bivalve samples [5]. The method used herein was developed for the simultaneous detection of chloramphenicol classes, quinone classes, MG, and LMG in shrimp samples. This is the most efficient and energy-conservative method for veterinary drug extraction. However, the same method could not be employed to analyze the nitrofuran metabolite residues in aquatic samples; because of their chemical structural characteristics, nitrofuran metabolites in food samples were extracted using 2-NBA for derivatization [19].

To validate the shrimp sample analysis method, as recommended by the TFDA [11], the acceptable recovery rate had to be 70–120% with RSD < 15% for chemical residues in food matrixes detected in the 0.1–1 mg/kg range; 70–120% with RSD < 20% for those detected in the 0.01–0.1 mg/kg range; 60%–125% with RSD < 30% for those detected in the 0.001–0.01 mg/kg range; and 50–125% with RSD < 35% for those detected within ≤0.001 mg/kg. According to our results, veterinary drug residues detected within ≤0.001 mg/kg and 0.001–0.01 mg/kg ranges demonstrated a recovery rate of 80–120% with an RSD of <10% and a recovery rate of 70–120% with an RSD of <15%, respectively. The TFDA also recommends various LOQs, including 0.3 ng/g in chloramphenicol; 5 ng/g in florfenicol and thiamphenicol; 10 ng/g in quinolone classes; 0.5 ng/g in MG and LMG; and 1 ng/g in nitrofuran metabolites, for aquatic food for the assessment of veterinary drug residues in seafoods [20]. Compared with the LOQs recommended by the TFDA, the LOQs obtained using our analytical method were lower and can be employed to detect trace veterinary drug residues. Therefore, the analytical methods employed herein conform to the recommendations of the TFDA.

The regulations entitled Tolerances for Residues of Veterinary Drugs in Food, established by the Ministry of Health and Welfare of Taiwan, state that nitrofuran metabolites, chloramphenicol, MG, and LMG are banned from use in shrimp culturing because of concerns that pertain to mutagenicity and carcinogenicity [5]. In addition, food-producing animals and products containing these chemicals exported by third-world countries are prohibited in Japan and the European Union, the major importers of Taiwanese marine products. Based on methodologies available for detecting banned compounds in edible products, the Department of Health of Taiwan [21] and EU Commission [22] have both established a maximum residual permissible limit (MRPL) of 1 ng/g for each nitrofuran metabolite in aquaculture, marine, and poultry meat products. Furthermore, the EU Commission stipulates an MRPL of 0.3 ng/g for chloramphenicol and 2 ng/g for MG plus LMG [16] in all food products of animal origin to ensure that customers worldwide are given the same level of protection. According to the aforementioned guidelines, the LOQs of our methods executed for identifying the levels of chloramphenicol, dye, and nitrofuran metabolite residues in shrimp meet the MRPL.

The chromatography–mass spectrometry screening of carcinogenic antimicrobials—such as nitrofuran metabolites, chloramphenicol, MG, and LMG—in 53 shrimp samples demonstrated a positive

result, with the chloramphenicol concentration being greater than the MRPL of chloramphenicol set by the EU Commission (0.31 ng/g in a grass shrimp sample). Administering chloramphenicol to food-producing animals is banned in Taiwan. Some aquaculture farmers use chloramphenicol regardless, however, because it is a broad-spectrum, inexpensive, and readily available antibiotic [23]. Mixed breeding has caused chloramphenicol to be employed for the prevention as well as treatment of infectious diseases in shrimp. In the present study, the proportion of positive identification of banned veterinary drugs was 1.89% (1/53). In the analysis of cultured shrimp in Bangladesh, the detection of chloramphenicol and nitrofuran metabolite residues revealed a violation ratio of 8.37% (118/1409) in 2008, 8.16% (182/2230) in 2009, and 5.81% (122/2098) in 2009 [24]. In addition, in the Canadian Total Diet Study from 1994 to 2004, the detection of MG, LMG, and nitrofuran metabolite residues revealed a violation ratio of 20.0% (6/30) [25]. In Ireland, exposure to nitrofuran residues was assessed from 2009 to 2010, which revealed a violation ratio of 5.68% (5/88) in the detection of SEM residues [26]. However, in the aforementioned reports, only two classes of nitrofuran metabolites and chloramphenicol or three classes of MG, LMG, and nitrofuran metabolites were detected. Our present findings differ from those of TFDA surveys. In reports in recent years, the violation ratio of banned veterinary drug residues in shrimp samples was 0% in 2013 (0/20) [27] and 0% in 2014 (0/20) [28]. These differences are partially accounted for by varying sample sizes. In addition, the samples collected in this study were obtained from shrimp production areas in Taiwan, whereas the samples collected by the TFDA may have been imported shrimp. Therefore, several categories of banned veterinary drugs in Taiwanese shrimp were appropriately detected in the present study.

In the present investigation, quinolone residues (3/53) were detected with a higher violation ratio than chloramphenicol (1/53) in aquaculture shrimp. Our study revealed that quinolones were the predominant compounds in the aquacultured shrimp samples in Taiwan, which was similar to the results of a survey conducted in China [29], Vietnam [30,31], and Thailand [31] following intensive use in aquaculture to treat bacterial infections, which polluted aquatic habitats and had adverse effects on the health of freshwater organisms. Quinolones were detected in 8.70% (2/23) and 7.14% (1/14) of whiteleg and giant river shrimp samples, respectively. In all shrimp samples, the predominant residue was flumequine at 3.77% (2/53), followed by enrofloxacin at 1.89% (1/53). The results of the present study are similar to those of the survey conducted by the TFDA. Compared with a report of the TFDA in 2012, the violation ratio of quinolone residues in shrimp samples was 4.0% in 25 samples, which was positive with flumequine at 21.0 ng/g in one shrimp sample [32]. From the data available, we concluded that flumequine continues to be used as a growth promoter and prophylactic agent in aquatic products because of its affordability and effectiveness. Other surveys in Asia [31,33] have reported that flumequine has been the most widely used synthetic antibiotic in aquaculture, especially because of its relative stability to resist bacterial degradation in water and sediments. In addition, flumequine residues were detected in trace amounts; only a concentration of 1.01–1.52 ng/g or higher triggers action by the TFDA (withdrawal of the product and issuance of an alert). The results of the surveys reviewed herein indicate that the Taiwanese population is exposed to trace amounts of flumequine that do not pose an immediate risk to health through the consumption of shrimp. Therefore, Taiwan's regulatory authorities and producers should continually monitor aquatic products and prevent sources of contamination, ensuring the chemical safety of commercially available foods.

Parameter guidelines indicate how the risk to organisms such as humans can be assessed by stipulating criteria related to the ADI, hazard quotients, provisional tolerable weekly intake, and excess cancer risk [6,34,35]. Guidelines for the ADI, such as those formulated by the FAO and WHO, facilitate the assessment of risks to organisms, including humans [6]. The ADI is a single value, however, and eating habit and consumption rate differences are not considered in its calculation [36,37]. The JECFA [38] and US EPA [39] have proposed a new and highly accurate measure for the estimation of chronic dietary intake: the EDI. In this study, we concluded that the ADI was not exceeded by the corresponding daily exposure. Because few residual quinolones were discovered, the estimated EDI revealed that consumption of the investigated shrimp would result in considerably less dietary

intake of enrofloxacin and flumequine in the Taiwanese population than that stipulated by the ADI. Furthermore, when assessed against the ADIs, the EDIs calculated in this study indicated no risk to health due to shrimp consumption. The EDIs were lower than 1% of the ADIs in this study, indicating negligible risk [6,38]. Thus, the levels of quinolone in Taiwanese food products can be concluded to not negatively affect health. Because of the potentially adverse effects of antibiotics on health and aquatic environments, the impact of these pollutants must be urgently evaluated further.

## 5. Conclusions

In the present study, we analyzed the residues of chloramphenicol, florfenicol, thiamphenicol, MG, LMG, nitrofuran metabolites, danofloxacin, difloxacin, enrofloxacin, flumequine, and sarafloxacinthe in shrimp samples; methods used were validated according to the EU criteria and complied with the MRPLs established by the EU and TFDA. The residues of banned veterinary drugs chloramphenicol and quinolone, with no MRL recommended, were detected in 53 shrimp samples. We observed that one shrimp sample contained chloramphenicol, one shrimp sample contained enrofloxacin, and two shrimp samples contained flumequine. Notably, only trace amounts of all residues were discovered, indicating no immediate risk to health because the EDIs were considerably lower than the FAO/WHO-defined ADIs. Enrofloxacin and flumequine contamination following shrimp consumption in Taiwan appears to present a negligible threat to human health. However, the concern regarding pharmaceuticals and their adverse effects on the environment and human health is increasing, and a background information system on the consumption of veterinary antibiotics through shrimp must be established and improved, thus providing a monitoring and management framework. The health and agricultural authorities can use the present study findings as a valuable reference when improving contaminant regulation in aquaculture.

**Supplementary Materials:** The following are available online at http://www.mdpi.com/2076-3417/9/12/2463/s1, Supplementary Table S1: Retention time and MS/MS fragmentation conditions for veterinary drugs and their corresponding internal standards, Supplementary Table S2: Recovery, repeatability, and LOQ of veterinary drugs spiked into whiteleg shrimp.

**Author Contributions:** M.-Y.T. and C.-F.L. conceived the idea and performed experiments. W.-C.Y., C.-T.L., and K.-H.H. assisted in recombinant construction. G.-R.C. wrote, reviewed, and edited the manuscript. M.-Y.T. and C.-F.L. contributed equally to this work.

**Funding:** This research received no external funding.

**Acknowledgments:** This study was supported by the Ministry of Science and Technology (Taiwan) (MOST 107-2313-B-415-012) and, in part, by the Taichung Veterans General Hospital (Taiwan) and National Chung-Hsing University (Taiwan) (TCVGH-NCHU-10776013). This manuscript was edited by Wallace Academic Editing.

**Conflicts of Interest:** The authors declare no conflict of interest.

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
