# Peer review of "Health Risk Assessment of Banned Veterinary Drugs and Quinolone Residues in Shrimp through Liquid Chromatography–Tandem Mass Spectrometry"

_applsci, doi:10.3390/app9122463_

Round 1
Reviewer 1 Report
This is a well-written manuscript that describes the analysis of veterinary drugs in Taiwanese aquaculture products. The analytical chemistry part of the manuscript is minimized in favor of data/result discussion. Within the scope of my LCMS expertise, I was able to double check the LC method and SRM/MRM transitions in the MS method and did not find any potential problems. I recommend publication as the data will be interesting for the risk assessment community.
Author Response
Response:
Thank you for your affirmation.
Revision:
No revision.
Reviewer 2 Report
Overall the paper needs a careful language revision.
Here are a few examples, but it needs a revision.
Eg:
Aquaculture in Taiwan has a long history of more than three centuries. It has rapidly expanded, diversified, intensified, and advanced technologically through spectacular times from the 1960s to the 1990s.
This has an inadvertent ecological impact and raises health concerns such as increased risk of allergies, carcinogen exposure and the development of bacteria resistant to antibiotics
I have some additional remarks:
Have you considered age of the shrimp analyzed, is there a correlation between how long the shrimp has grown and the exposure to the drugs? How will that affect the results?
Author Response
Comments and Suggestions for Authors
Overall the paper needs a careful language revision.
Here are a few examples, but it needs a revision.
Eg:
Aquaculture in Taiwan has a long history of more than three centuries. It has rapidly expanded, diversified, intensified, and advanced technologically through spectacular times from the 1960s to the 1990s.
This has an inadvertent ecological impact and raises health concerns such as increased risk of allergies, carcinogen exposure and the development of bacteria resistant to antibiotics
Response:
Our manuscript was revised and edited by a native English speaker from a professional academic editing service before the first submission. In addition, the present modified version has also been re-edited according to the editors’ suggestion with the similarity index. Please refer to our attachment—“English Editing Certificate.”
Revision:
Additional revisions have been made from your suggestion (page 1, line 42; page 2, line 60)
“1. Introduction
Taiwan has a geographic location and environment conducive to aquaculture development. Aquaculture in Taiwan has a long history of more than three centuries, and it has rapidly expanded, diversified, intensified, and technologically advanced from 1960 to the 1990s [1]. Despite Taiwan’s land and water resource limitations, it is one of the major aquaculture producers in the world; therefore, it was once called the “kingdom of aquaculture” [2]. …”
“…However, when such drugs are heavily employed in aquaculture, aquaculture products may contain drug residues, potentially exposing the consumers of the products to these residues. This has an inadvertent ecological impact and raises health concerns such as increased risk of allergies, carcinogen exposure and the development of bacteria resistant to antibiotics [5]. Because of their benthic feeding behavior, shrimp can be used to indicate the levels of chemicals in aquatic environments [7]. Analyzing the amounts of pollutants in shrimp indicates the environmental levels of veterinary drugs and the extent to which drugs are transferred through the trophic chain.”
Additional revisions have been made.
I have some additional remarks:
Have you considered age of the shrimp analyzed, is there a correlation between how long the shrimp has grown and the exposure to the drugs? How will that affect the results?
Response:
Based on these considerations for health risk assessment, our shrimp samples were collected at the weight of one sample was approximately 12 g. The size and weight of shrimps are cultivated for one year in land-based ponds. Moreover, this type of shrimp could be marketed and edible for most Taiwanese. Therefore, we did not study the residue effects between the age and residual drugs of the shrimps. Thanks for your suggestion and we will continue to study this correlation in our future work.
Revision:
To condense the manuscript and shorten the revised version, this part was not integrated into the revision.
Thank you for your suggestions for revisions. We have carefully considered all comments and have made appropriate changes to the manuscript.
